# Prehabilitation to improve outcomes afteR Autologous sTem cEll transplantation (PIRATE): A pilot randomised controlled trial protocol

**Amy M. Dennett** [1,2]*, **Judi Porter** [3], **Stephen B. Ting** [4,5], **Nicholas F. Taylor** [1,2]

**1** School of Allied Health, Human Services and Sport, La Trobe University, Bundoora, Australia, **2** Allied Health Clinical Research Office, Eastern Health, Box Hill, Australia, **3** Institute for Physical Activity and Nutrition (IPAN), School of Exercise and Nutrition Sciences, Deakin University, Burwood, Australia, **4** Department of Clinical Haematology, Eastern Health, Box Hill, Australia, **5** Eastern Health Clinical School, Monash University, Box Hill, Australia

* amy.dennett@easternhealth.org.au

**Data Availability Statement:** No datasets were generated or analysed during the current study. All relevant data from this study will be made available upon study completion.

## Abstract

### Background

Autologous stem cell transplant is a common procedure for people with haematological malignancies. While effective at improving survival, autologous stem cell transplant recipients may have a lengthy hospital admission and experience debilitating side-effects such as fatigue, pain and deconditioning that may prolong recovery. Prehabilitation comprising exercise and nutrition intervention before stem cell transplant aims to optimise physical capacity before the procedure to enhance functional recovery after transplant. However, few studies have evaluated prehabilitation in this setting. We aim to explore preliminary efficacy of improving physical capacity of prehabilitation for people undergoing autologous stem cell transplant.

### Methods

The PIRATE study is a single-blinded, parallel two-armed pilot randomised trial of multidisciplinary prehabilitation delivered prior to autologous stem cell transplantation. Twenty-two patients with haematological malignancy waitlisted for transplant will be recruited from a tertiary haematology unit. The intervention will include up to 8 weeks of twice-weekly, supervised tailored exercise and fortnightly nutrition education delivered via phone, in the lead up to autologous stem cell transplant. Blinded assessments will be completed at week 13, approximately 4 weeks after transplant and health service measures collected at week 25 approximately 12 weeks after transplant. The primary outcome is to assess changes in physical capacity using the 6-minute walk test. Secondary measures are time to engraftment, C-reactive protein, physical activity (accelerometer), grip strength, health-related quality of life (EORTC QLQ-C30 and HDC29 supplement), self-efficacy and recording of adverse events. Health service data including hospital length of stay, hospital readmissions,

**Funding:** AD received an Eastern Health Foundation grant to support this work. They will not play any role in the study design, data collection and analysis, or publication process https://www.easternhealth.org.au/foundation/

**Competing interests:** The authors have declared that no competing interests exist.

emergency department presentations and urgent symptom clinic presentation at will also be recorded.

## Discussion

This trial will inform design of a future definitive randomised controlled trial and implementation of prehabilitation for people receiving autologous stem cell transplant by providing data on efficacy and safety.

## Trial registration

The PIRATE Trial has been approved by the Eastern Health Human Research Ethics Committee (E20/003/61055) and is funded by the Eastern Health Foundation. This trial is registered with the Australian New Zealand Clinical Trials Registry ACTRN12620000496910. Registered April 20, 2020.

## Background

Autologous stem cell transplantation is an effective procedure that provides long-term disease control and is standard care for many haematological malignancies including multiple myeloma, lymphoma and less frequently leukaemia [1]. In selected diseases, the survival rate of patients receiving autologous stem cell transplant is up to 94% at 10 years [2]. Autologous stem cell transplants are increasingly available to patients with haematological malignancies, with over 36,000 autologous stem cell transplants completed globally each year [3].

While survival from autologous stem cell transplant is high, this procedure places patients at risk of significant short and long-term adverse effects. Patients preparing for autologous stem cell transplant receive high doses of chemotherapy and radiotherapy to eliminate tumor cells from the body. This results in a period of immunosuppression whereby people are at risk of severe infections, prolonged bed rest and immobility during a lengthy hospitalisation while awaiting bone marrow stem cell recovery. This intensive treatment frequently results in a myriad of toxic effects including sepsis, fatigue, deconditioning, peripheral neuropathy, and psychological distress causing subsequent functional decline and poor quality of life [4]. These problems may persist long after transplant with 35% of long-term transplant recipients experiencing fatigue [5]. Persistent muscle weakness, pain, cognitive impairment and distress are also common [6]. Autologous transplant recipients also have shorter life expectancy and higher risk of secondary cancers than the general population and experience a high rate of hospital readmission [2, 7].

Exercise-based rehabilitation plays an important role in mitigating the negative effects of cancer and its treatment. Guidelines recommend people with cancer participate in three times weekly aerobic and twice-weekly resistance exercise training to improve health outcomes [8]. Exercise training reduces fatigue, improves strength and quality of life in people who have received autologous stem cell transplant [9, 10]. However, it is unknown how best to deliver exercise-based rehabilitation for people receiving autologous stem cell transplant.

Supervised exercise yields the greatest benefits for cancer survivors [8]. However, the majority of previous trials for people receiving autologous stem cell transplant have provided unsupervised training [9, 10]. The optimal time period to provide exercise-based rehabilitation is also unknown, with most trials completed to date targeting the peri and post-transplant

period [9, 10]. One trial found an 18-week supervised exercise-rehabilitation program after transplant was not cost-effective [11]. Preliminary evidence from 10 trials including both allogeneic and autologous stem cell transplant recipients participating in exercise across the treatment continuum suggests that training prior to stem cell transplant may be superior to post-transplant [10].

The addition of nutrition interventions to exercise may further enhance outcomes after autologous stem cell transplant as this group has complex nutrition needs [12]. Nutritional outcomes after transplant include poor appetite, mucositis and other gastrointestinal complications, which may lead to malnutrition [13]. Nutrition interventions provided before and after autologous stem cell transplant reduce weight-loss and hospital length of stay [14–17]. A multidisciplinary approach comprising exercise and nutrition could further optimise recovery from autologous stem cell transplant as shown in previous studies combining nutrition and exercise interventions during prehabilitation for other cancers [18].

The period before autologous stem cell transplant may be a critical period to intervene with rehabilitation (i.e. prehabilitation). Prior to stem cell transplant, patients already experience impaired physical function and quality of life [19]. Consequently, patients with haematological malignancies face many barriers to participating in physical activity and want to increase their physical activity behaviors [20]. Given functional performance status is positively associated with survival after transplant [21] patients preparing for transplant need to achieve a minimum level of function (Karnofsky Performance Status ≥70 or Eastern Cooperative Oncology Group score ≤2) [1]. Prehabilitation aims to build functional reserve to prepare people to cope with the physical and psychological demands of treatment. It also represents a 'teachable moment' whereby patients can gain skills and increase their self-efficacy to be physically active in the long-term [22, 23]. Systematic reviews have shown prehabilitation to be effective in people with other cancers receiving surgery [18]. However, limited high-quality studies have been completed for people with haematological cancer. One non-randomised feasibility study including 29 participants provided evidence a 4 to 6-week supervised exercise-rehabilitation program was safe and well accepted by patients preparing for autologous stem cell transplant [24]. Patients receiving prehabilitation required a shorter length of stay and earlier recovery of blood counts than the control group [24]. Previous studies on prehabilitation for autologous stem cell transplant have only focused on exercise, and did not consider other interventions such as nutrition support [18]. No randomized trials have evaluated the impact of supervised, multidisciplinary prehabilitation for people preparing for autologous stem cell transplant.

## Aims and objectives

The aim of this pragmatic pilot trial is to explore whether supervised, multidisciplinary prehabilitation for patients preparing for autologous stem cell transplant can positively affect physical capacity after stem cell transplant. In addition to physical capacity, secondary efficacy outcomes will include time to engraftment, C-reactive protein levels, objective physical activity levels, grip strength, nutritional status, health-related quality of life and self-efficacy. Adverse events will also be reported as an indicator of safety. This trial will establish estimates of effect to inform a larger definitive trial. The trial will be reported consistent with the CONSORT statement for pilot and feasibility trials [25].

## Methods

### Study design

This is a prospective, parallel, single-blind, pragmatic, pilot randomised controlled trial to assess preliminary efficacy of multidisciplinary prehabilitation on post-autologous stem cell

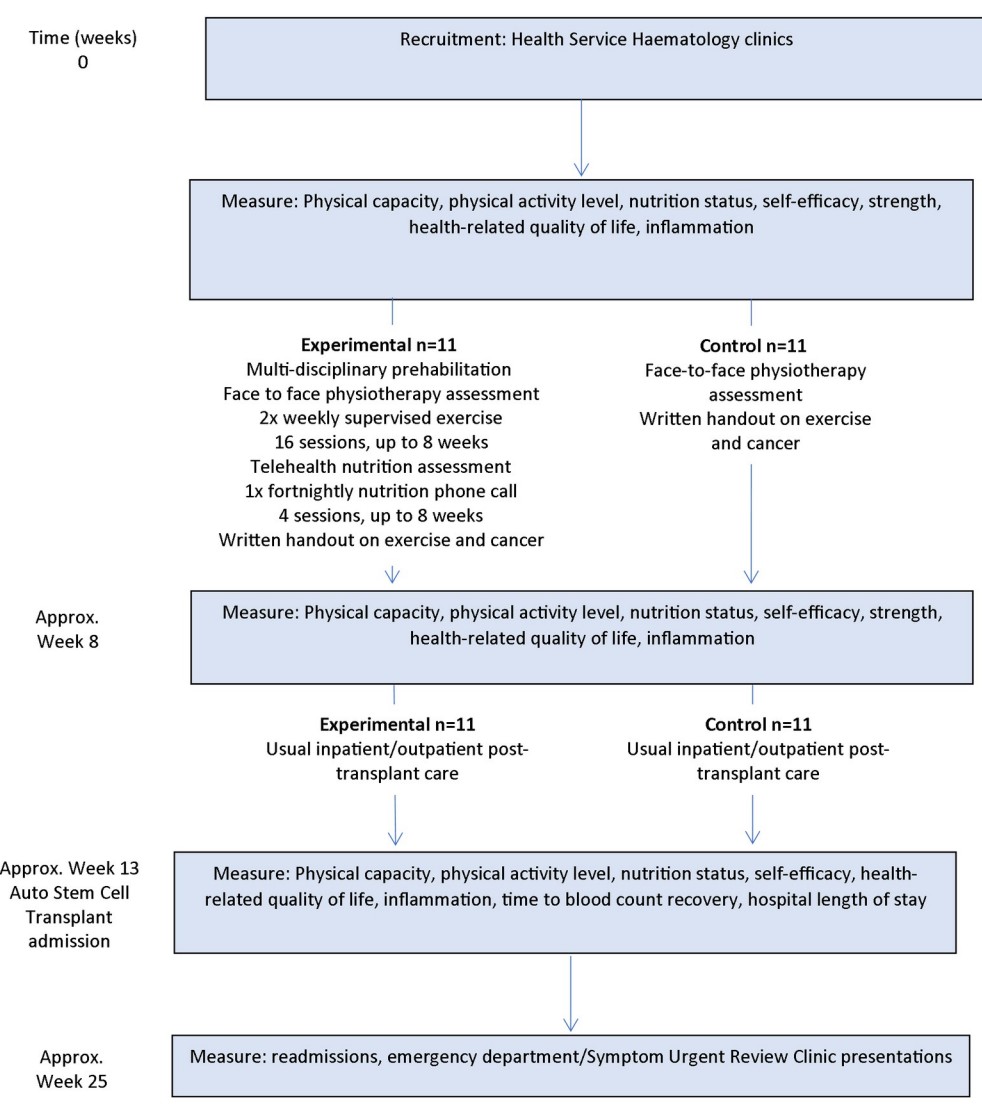

**Fig 1. Trial design.**

transplant physical capacity. An intervention of up to 8 weeks will be completed pre-transplant with participants followed up at week 13, approximately 4 weeks after transplant and health service measures collected at week 25, approximately 12 weeks after transplant (Fig 1 and S1 Fig). This trial has been approved by the hospital's ethics committee (E20/003/61055).

## Randomisation procedures

Eligible participants who have completed baseline measurements will be randomly allocated to the prehabilitation group or usual care control group according to an online computer-generated randomisation program, www.randomization.com, using permuted blocks without stratification. Allocations will be prepared prior to trial commencement by an independent researcher with no role in subject recruitment or administration of trial interventions. Participants will be allocated by the trial coordinator after baseline assessment by contacting the independent researcher by email for random group allocation.

## Setting

The trial will be conducted in a public, tertiary cancer treatment unit that conducts approximately 30 autologous stem cell transplants annually.

## Patient selection and consent

All patients referred to the haematology cancer unit for autologous stem cell transplant will be approached to participate in the trial. Potential candidates will be advised about the trial by clinic staff verbally and/or through flyers. If a patient gives permission to being contacted about the research project, they will be contacted by a member of the research team who will provide them with details of the study and arrange an outpatient appointment to provide an opportunity for questions to be clarified and to provide written informed consent.

## Inclusion and exclusion criteria

Participants will be eligible if they are aged 18 years and over; have a haematological malignancy and are waitlisted for autologous stem cell transplant; and are able to give written informed consent.

Participants will be excluded if they: are medically unfit to participate in exercise as determined by a physiotherapist and/ or medical practitioner based on published recommendations [26]; have low physical performance status (Australian-modified Karnofsky Performance status (AKPS) of <60 or Eastern Cooperative Oncology Group (ECOG) score >2); or have cognitive impairment precluding ability to provide written, informed consent as assessed by their treating clinician.

## Intervention

All participants, whether allocated to experimental or control groups will receive usual care. Usual care will include an initial assessment with a research physiotherapist who, at this session will also provide standardised written instructions with guidelines for exercise after cancer [27–29] and a referral to a sub-acute oncology rehabilitation program after stem cell transplant. They will continue to receive their usual medical care, which may include adjuvant chemotherapy, radiotherapy, inpatient admission post-transplant, specialist, nursing and allied health outpatient appointments and visits to their general practitioner.

## Prehabilitation intervention

Following physiotherapy assessment and randomisation, participants allocated to the experimental group will have an individualised exercise program designed for them based on assessment findings and goals. In a deviation from initial protocol due to ongoing COVID-19 restrictions, a home-based intervention will be implemented. Participants will undertake twice-weekly, 60-minute exercise sessions individually for up to 8 weeks pre-transplant following a 1-week period of physical activity monitoring (Table 1). Sessions will be individually tailored and supervised by a physiotherapist with cancer rehabilitation experience. They will comprise aerobic and resistance exercise, completed at a moderate intensity (4–6 BORG rating of perceived exertion (RPE) and/or 60–80% Heart rate (HR) maximum for aerobic exercises, and 10–12 repetition maximum (RM) for resistance exercise in accordance with guidelines using a combination of hospital-owned and patient's own exercise equipment. Participants will aim to aim to complete 20–25 minutes aerobic exercise, 20–25 minutes of resistance exercise, 5-minutes of flexibility or balance training (as indicated) and 5-minute warm up and cool down. Exercise intensity during aerobic exercise will be monitored by the physiotherapist

**Table 1. Intervention description using the template for description and replication checklist (TIDier).**

| | Experimental Group | | Control Group |
|---|---|---|---|
| **Brief Name** | Prehabilitation | | Usual Care |
| **Why** | Prehabilitation may build functional reserves to better cope with transplant | | Pragmatic trial design |
| **What: Materials** | • Participants will use combination of hospital-owned and their own exercise equipment:<br>• Free weights<br>• Resistance exercise bands<br>• Portable stepper<br>• Participants will receive 1) standardised written handout exercise 2) standardised written handout nutrition 3) referral to Oncology Rehabilitation Program post-transplant<br>• Participants will receive a Fitbit Inspire device worn continuously at the wrist for the duration of their prehabilitation period<br>• Participants will receive usual hospital care | | • Participants will receive 1) standardised written handout exercise 2) referral to Oncology Rehabilitation Program post-transplant<br>• Usual care also includes usual medical care which may include adjuvant chemotherapy, radiotherapy, inpatient admission post-transplant, specialist, nursing and allied health outpatient appointments, visits to their general practitioner and general advice from their medical team to remain active and eat a healthy diet. |
| **What Procedures** | | | |
| **Provider** | Physiotherapist and dietitian with oncology experience provided by the hospital | | Usual hospital staff |
| **How** | Face to face sessions +/- telehealth (pending COVID-19 lockdown directives) | | No intervention |
| **Where** | Patient's home | | No intervention |
| **When/How much Intensity Frequency Session time Overall duration** | **Exercise** | *Nutrition* | • Standardised written advice about exercise and cancer guidelines |
| | Moderate (BORG 4–6) 60–80% heart rate maximum 10–12 repetition maximum | | |
| | 2x weekly supervised 1X weekly unsupervised | Fortnightly | |
| | 60 minutes | 30 minutes | |
| | Up to 8 weeks | | |
| **Tailoring** | • Individualised exercise program and nutrition advice based on initial consultation and goals | | • None |
| **Trial fidelity** | • Staff with a background in oncology physiotherapy and dietetics who had prior formal training were employed by the hospital to provide the intervention<br>• Exercise log-books will be completed and reviewed by research staff.<br>• Nutrition log-books will be completed and reviewed by research staff.<br>• Records of the number and duration of completed sessions.<br>• Monthly meetings with clinical research staff | | • Participants will be asked if they participated in any physical activity or nutrition intervention during the usual care period. |

using the modified BORG scale and a Fitbit device. During weeks 1 and 2, participants will aim to exercise at a BORG RPE of 3 (moderate), and by week 8 participants will aim to exercise at a 5–6 (hard) on the scale. For resistance exercise, weights will be progressed once a participant is achieving 2 to 3 sets of 10–12 repetitions. Resistance exercise may include upper and lower body resistance exercise such as squats, step ups, free weights, wall push-ups, resistance exercise bands. Aerobic exercise may include walking, stationary cycle, or portable stepper machine. Various upper and lower body stretches and balance exercise will be incorporated into the program as required. Participants in the prehabilitation group will be encouraged to complete an additional once-weekly 30-minute aerobic exercise training session independently and will be provided with a Fitbit Inspire device and collaborate with the physiotherapist to set and review a daily steps goal to further assist compliance with exercise guidelines for people with cancer [8]. Patients will also be instructed to remain as active as possible and avoid prolonged periods of sitting and lying in the period following stem cell transplant.

A dietitian will provide written information and fortnightly phone or video calls over the 8-week period (up to 4 sessions) to patients offering tailored medical nutrition therapy based on initial dietetic consultation. The focus will be on supporting oral intake and maintaining nutritional status during the prehabilitation period. Guidelines for managing potential gastro-intestinal symptoms during the post-transplantation period will also be provided [13].

The fidelity of the intervention will be monitored by recording the content of exercise and nutrition sessions in logbooks. The exercise log will record exercise type, intensity, duration, frequency and modifications. The nutrition log will record weight, general health changes, 24-hour food recall and session goals. Number and duration of completed sessions with the physiotherapist and dietitian will also be recorded and monthly meetings will be held with clinical staff.

## Control group

Participants randomised to the control group will receive their usual medical care and receive standardised written instructions with guidelines for exercise after cancer from the physiotherapist [27–29].

## Study outcomes

Participants will complete an assessment of physical capacity, physical activity, inflammation, time to engraftment, health-related quality of life, self-efficacy, nutritional status and muscle strength at baseline and after the intervention phase at approximately week 8 (pre-transplant) and week 13 (post-transplant infusion). Hospital length of stay will be recorded at week 13, and emergency department, Symptom and Urgent Review Clinic (SURC) presentations and hospital readmissions will be recorded at week 25 from hospital data bases. A trained allied health clinician blind to group allocation will complete baseline and follow-up assessments to ensure blinding of outcome measures. Peripheral blood count analysis and blood product use will be completed by an independent assessor, blinded to group allocation. Primary and secondary outcomes are outlined in Table 2 and S2 File.

Adverse events related to the intervention as defined by the World Health Organization [30] will be documented to report safety of the intervention. The event may or may not be related to the intervention, but it occurs while the person is participating in the intervention phase (during prehabilitation) of the trial. Adverse events will be categorised as minor adverse events or serious adverse events. A minor adverse event is defined as an incident that occurs while the person is participating in the intervention that results in no injury or minor injury (e.g. fatigue, exacerbation of pre-existing musculoskeletal pain) that requires none or minor

**Table 2. Primary and secondary outcomes.**

| Primary outcome | Measure/source | Definition |
|---|---|---|
| Physical Capacity | 6-Minute Walk Test | Change in walk distance (m) pre-post intervention. Primary endpoint is 4 weeks post-transplant infusion |
| **Secondary outcomes** | | |
| Physical Activity | ActivPal[TM] | Change in time spent walking, standing, sitting, sit-to-stand transitions, and step count pre and post intervention. Participants will wear the activity monitor continuously or 8 consecutive days. Only complete 24-hour recording days will be included for analysis. However, as monitors may need to be removed for the purpose of swimming or bathing, evidence of non-wear matching with an activity logbook will still be included. |
| Health-Related Quality of Life | EORTC-QLQ C30 and EORTC QLQ-HDC29 | Change of score on validated quality of life questionnaires QLQ-C30 and HDC29 pre and post intervention. |
| Self-efficacy for physical activity | Questionnaire developed using HAPA (Additional file 1) | Change of score on self-efficacy questionnaire for physical activity pre and post intervention. |
| Nutritional status | PG-SGA | Change of score on validated PG-SGA pre and post intervention. |
| Handgrip strength | Jamar handgrip dynamometer | Change in handgrip strength (kg) pre and post intervention assessed using the best measure of 6 trials (3 in each hand). |
| Inflammation | C-Reactive Protein | Change in CRP levels pre and post intervention. Patients will be instructed not to undertake moderate to vigorous intensity exercise for 24 hours prior to collection. |
| Stem cell engraftment | Routine blood samples | Number of days from transplant to engraftment. Engraftment is defined as neutrophils $>0.5 \times 10^9/L$ for three days without support and platelets $>50 \times 10^9/L$ for five days without transfusion. |
| Hospital length of stay | Hospital database | Days that the patient is in the hospital from day of stem cell infusion to day of discharge. |
| ED/SURC presentations | Hospital database and electronic medical record | Number of emergency department presentations and Symptom and Urgent Review Clinic (SURC) presentations over three months after discharge from the autologous stem cell transplant admission. |
| Hospital re-admissions | Hospital database | Number of hospital readmissions over 3 months after discharge from the autologous stem cell transplant admission and associated inpatient days with each readmission |

EORTC QLQ: European Organisation for Research and Treatment of Cancer Quality-of-life Questionnaire; PG-SGA Patient-Generated Subjective Global Assessment; HAPA: Health Action Process Approach

medical intervention. A serious adverse event is defined as an incident that occurs while the person is participating in the intervention that results in death, serious injury or re-hospitalisation. Reasons for non-participation in an exercise session or non-completion of the program, including both medical (e.g. pain, fatigue, unwell) and psychosocial reasons (e.g. work-related, forgot, other appointments), will be recorded. Complications related to the stem cell transplant procedure will also reported for each group (e.g. infection, bleeding, mucositis, parental nutrition requirements, intensive care support).

Other routinely collected data will be used to describe the sample including age, gender, cancer type, cancer stage, treatment regimens, co-morbidities, functional performance status (AKPS and ECOG), body mass index.

## Sample size estimation

No minimal clinically significant difference has been calculated in patients receiving autologous stem cell transplant therefore it was estimated to be 41 m, based on half a standard deviation [31] of scores of a mixed cohort of cancer survivors [32]. For this pilot study, to produce a one-sided 80% confidence limit that would exclude an effect of 0.5 would require a sample size of n = 12; and a one-sided 90% confidence interval would require a sample size of n = 28. Therefore, we aimed to sample n = 22 for this pilot study [33]. Approximately 30 people are treated with autologous stem cell transplant at the health service annually. Our sample size represents a recruitment rate of 75% which is similar to a recently completed cancer rehabilitation trial at the health service [34].

## Statistical analysis

The primary outcome (physical capacity at 4-weeks post-transplant) will be analysed using linear mixed effects models. Modelling will account for variation in baseline values. This method accounts for within-participant dependence of observations over time, and for missing data, allowing some participants to have missing observations at certain time points. If more than 5% of data are missing, a multiple imputation process will be used, providing the assumption data are missing at random is met. A similar approach (linear mixed effects model) will be used for analysis of continuous secondary outcomes collected longitudinally. As this trial is exploratory and there is no universal hypothesis our analysis will not correct for multiplicity [35]. The time spent in moderate to vigorous physical activity will be estimated using a cut-off of 100 steps/minute for moderate intensity physical activity [36]. The proportion of participants meeting physical activity guidelines will be described and compared between groups with a risk ratio. The number of emergency department, SURC presentations and hospital admissions will be reported as an incidence rate ratio using a negative binomial regression model. To avoid bias and to maximize the randomisation process, all available data will be analysed according to allocation (intention to treat analysis), regardless of compliance. Data will be analysed using IBM SPSS version 28.

## Discussion

This trial aims to capitalise on the 'teachable moment' of cancer diagnosis to improve physical capacity and expedite the recovery of patients undergoing autologous stem cell transplant. The benefits of rehabilitation for people with cancer, including exercise and nutrition interventions, are well documented. However, rehabilitation interventions are often not considered until treatment completion [37] at which stage impairment and further co-morbidity has developed. Early intervention provided by prehabilitation may mitigate the extensive treatment burden of autologous stem cell transplant and better prepare people physically and mentally for their lengthy recovery. If this trial finds preliminary evidence of efficacy and safety it may inform design of a definitive randomised controlled trial that may transform the stem cell transplant clinical pathway.

A strength of this trial is the inclusion of health service outcomes such as length of stay, emergency department presentations and hospital readmissions. Access to multidisciplinary cancer rehabilitation services, including prehabilitation, is poor [38]. Results relating to these endpoints which are valued by hospital administrators and policy makers may help drive future implementation of prehabilitation in health services to improve patient access. Preliminary evidence suggests that rehabilitation interventions conducted in early phases of cancer treatment may deliver cost-savings. For example, trials of prehabilitation for people receiving lung cancer surgery demonstrate reductions in costly hospital stays of 4 days and complications by 67% [39]. A trial of rehabilitation during chemotherapy for women with breast cancer also demonstrated cost savings [40]. In comparison, a trial of exercise *after* autologous stem cell transplant was not cost effective [11]. Data relating to health service benefits will guide decision making to see prehabilitation translated to practice.

Another strength is the multidisciplinary nature of the trial. Two trials of prehabilitation for autologous stem cell transplant have recently been registered [41, 42] however, these both focus on exercise alone. As people undergoing autologous stem cell transplant are at high risk of malnutrition and loss of muscle mass, the inclusion of nutrition intervention is of upmost importance. A recent systematic review identified five studies where prehabilitation included a nutrition intervention prior to surgery for cancer [18]. Three of these studies included nutrition only interventions [43–45], while a combined exercise/nutrition intervention was integral

to two studies [43, 46]. Nutrition counselling in addition to a range of interventions (e.g. the addition of supplemental arginine, whey protein and other dietary modifications) were the focus of the intervention groups. Significant improvements in functional measures arose from these nutrition interventions, including in physical functioning and post-operative symptoms. The value of multimodal prehabilitation has been recognised in a surgical context [18, 47] and this could extend to pre-transplant cohorts. The sample size for this pilot study is sufficient to estimate whether the assumed clinically important effect of 0.5 is realistic. If we find the estimate to be greater than zero, this may justify proceeding to a fully-powered study [33].

One possible limitation of this trial is the inclusion of face-to-face therapy. This trial was approved and registered prior to the full realisation of the COVID-19 pandemic. The safety risks of face to face rehabilitation associated with the pandemic, especially in this particularly vulnerable group has led to a surge in telehealth to enable critical supportive care delivered remotely. Cancer survivors are accepting of telehealth describing it as convenient, reassuring and minimising treatment burden [48]. Early trials of telehealth for delivering rehabilitation interventions to cancer survivors are also promising, but have been largely limited to education and coaching delivered via phone [49, 50]. Oncology rehabilitation trials and programs around the world have had to alter their protocols to include novel solutions such as video conferencing, digital exercise programs and remote monitoring to overcome the challenges associated with COVID-19 [51]. In some parts of Australia, there has been some easing of COVID restrictions allowing community-based exercise. However, should the need arise, important lessons have been learnt in adapting to telehealth that can be adopted for this trial. Results may also not be generalisable to all settings that conduct autologous stem cell transplant. However, it is acknowledged this is a pilot and results will inform future robust trials on a larger scale.

Prehabilitation is an important area of oncology supportive care research that has potential to challenge current care delivery. This trial may provide preliminary evidence about an intervention that may optimise patient and health service outcomes and aid efforts to see prehabilitation integrated as standard care for people with haematological malignancies.

## Supporting information

**S1 Checklist. SPIRIT checklist.**
(DOC)

**S1 Fig. SPIRIT figure.**
(TIF)

**S1 File. Self-efficacy for physical activity questionnaire.** This file is a questionnaire developed by the researchers that will be used to assess self-efficacy for physical activity using the Health Action Process Approach.
(DOCX)

**S2 File. Rationale and psychometric properties of outcome measures.**
(DOCX)

**S1 Data.**
(DOCX)

## Acknowledgments

We would like to acknowledge the haematology and oncology rehabilitation staff at Eastern Health who will be supporting recruitment for this trial.

## Author Contributions

**Conceptualization:** Amy M. Dennett, Judi Porter, Stephen B. Ting, Nicholas F. Taylor.

**Data curation:** Amy M. Dennett.

**Formal analysis:** Amy M. Dennett.

**Funding acquisition:** Amy M. Dennett, Judi Porter, Stephen B. Ting, Nicholas F. Taylor.

**Investigation:** Amy M. Dennett.

**Methodology:** Amy M. Dennett, Judi Porter, Stephen B. Ting, Nicholas F. Taylor.

**Project administration:** Amy M. Dennett.

**Resources:** Amy M. Dennett.

**Supervision:** Judi Porter, Stephen B. Ting, Nicholas F. Taylor.

**Writing – original draft:** Amy M. Dennett.

**Writing – review & editing:** Amy M. Dennett, Judi Porter, Stephen B. Ting, Nicholas F. Taylor.

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
