## [Decision Letter · Decision Letter 0]

8 Jul 2022

PONE-D-21-32897Prehabilitation to Improve outcomes afteR Autologous sTem cEll transplantation (PIRATE): A pilot randomised controlled trial protocolPLOS ONE

Dear Dr. Dennett,

Thank you for submitting your manuscript to PLOS ONE. After careful consideration, we feel that it has merit but does not fully meet PLOS ONE’s publication criteria as it currently stands. Therefore, we invite you to submit a revised version of the manuscript that addresses the points raised during the review process.

Your manuscript has been assessed by two expert reviewers, whose comments are appended below. The reviewers have highlighted serious concerns about several aspects of the methodology and statistical analysis which must be addressed before your manuscript can be published in PLOS ONE. Please ensure you respond to each point carefully in your response to reviewers document, and modify your manuscript accordingly.

We look forward to receiving your revised manuscript.

Kind regards,

Joseph Donlan

Editorial Office

PLOS ONE

Journal Requirements:

3. Please include a separate caption for each figure in your manuscript

Reviewers' comments:

Reviewer's Responses to Questions

**Comments to the Author**

1. Does the manuscript provide a valid rationale for the proposed study, with clearly identified and justified research questions?

Reviewer #1: Yes

Reviewer #2: Yes

2. Is the protocol technically sound and planned in a manner that will lead to a meaningful outcome and allow testing the stated hypotheses?

Reviewer #1: Yes

Reviewer #2: No

3. Is the methodology feasible and described in sufficient detail to allow the work to be replicable?

Reviewer #1: No

Reviewer #2: No

4. Have the authors described where all data underlying the findings will be made available when the study is complete?

Reviewer #1: No

Reviewer #2: Yes

5. Is the manuscript presented in an intelligible fashion and written in standard English?

Reviewer #1: Yes

Reviewer #2: Yes

6. Review Comments to the Author

You may also provide optional suggestions and comments to authors that they might find helpful in planning their study.

Reviewer #1: This manuscript is very well-written. I would like to congratulate the authors for proposing this interesting topic. I only have a few minor concerns:

1) The authors mentioned that this is a “pilot randomized trial” in the title. However, I do not think this is a pilot study since the interventions and outcomes are well-defined. They might need to have a second look or provide a clear rationale and justifications for their methodology choice.

2) It is not clear to me why the authors decided to measure self-efficacy. I think they need to explain why it is important for this population. The short definition presented in table 2 is not enough to convince the reader of its relevance in this study. They would need to discuss self-efficacy and other outcomes somewhere in the methods section or maybe in the introduction.

3) The authors will do a great job recording reasons for non-participation in exercise (or non-adherence to exercise program) which is very important. However, they are planning to report only medical-related reasons, and my question is, why did they ignore psychosocial reasons for non-adherence? I think self-efficacy is also can be a reason for non-adherence as well. It is unnecessary (and not feasible) to report all reasons for non-adherence, but they need to discuss these issues since this is a study protocol paper. They need to be clear about this, and if they are interested only in medical reasons for non-completion or non-adherence, they should mention this explicitly in the methods section.

4) Finally, the validity of the outcome measures should also be discussed in the methods.

I am very excited to see the results of your study. Keep it up, and I wish you all the best,

Reviewer #2: The primary objective of this single-blinded, parallel two-armed pilot randomized trial is to determine preliminary efficacy of improving physical capacity of prehabilitation for people undergoing autologous stem cell transplant. The primary outcome is to assess changes in physical capacity using the 6-minute walk test. The secondary otucomes are (1) time to engraftment, (2) C-reactive protein, (3) physical activity (accelerometer), (4) grip strength, (5) health-related quality of life (EORTC QLQ-C30 and HDC29 supplement), (6) self-efficacy, and (7) recording of adverse events. Although this protocol has covered key components, there are several major concerns about the experimental design and data analysis plan of the trial.

Major critiques:

1. The overall goal of this pilot study should be the hypothesis generating instead of hypothesis testing.

2. The permuted block randomization method was proposed; however, it is unclear if the randomization process includes any stratification factors.

3. The power analysis is based on the effect size method, which is not very informative. The authors should perform the precision analysis, i.e., the estimated width of the 95% CI, to reassure the proposed study sample size is sufficient. In addition, please specify the statistical test that was used for the power analysis, and clearly specify if the power analysis was based on the one- or two-sided type I error.

4. The trial has one primary endpoint, but there are almost 10 secondary outcomes in this very small trial. The authors may need to apply the multiple comparison method for the secondary outcomes when reporting the results.

5. The discussion of the patients’ compliance measurement is superficial.

6. The authors should clearly discuss the statistical test that will be used for each secondary outcome.

7. The authors should clearly specify the methods that will be used to estimate the 95% Confidence Intervals (CIs).

8. The authors should discuss the statistical software that will be used to analyze the data.

7. PLOS authors have the option to publish the peer review history of their article (what does this mean?). If published, this will include your full peer review and any attached files.

Reviewer #1: **Yes: **Alhadi M. Jahan

Reviewer #2: No

---

## [Author Response · Author response to Decision Letter 0]

21 Jul 2022

We thank the reviewers for the opportunity to revise our manuscript and for their insightful comments. Please see below point by point response to the reviewer comments which we think have helped strengthen the manuscript. 

Reviewer #1: 

1) Comment: The authors mentioned that this is a “pilot randomized trial” in the title. However, I do not think this is a pilot study since the interventions and outcomes are well-defined. They might need to have a second look or provide a clear rationale and justifications for their methodology choice.

Response: We understand the reviewers concerns with referring to this trial as a pilot. However, while the interventions and outcomes are well defined the effect of the interventions on the outcomes are not. No studies have previously conducted a trial of supervised, multidisciplinary prehabilitation for people recovering from autologous stem cell transplant. Therefore, we do not know what effect our intervention will have on our primary and secondary outcomes. Our study is underpowered with sample size estimate based on a large effect size of 1.3. However, the lower bands of the 95% confidence intervals will provide a conservative estimate to inform a larger efficacy trial. Moreover, as per our response to reviewer 2, comment 3, the sample size for this pilot study is sufficient to estimate whether the assumed clinically important effect of 0.5 is realistic. If we find the estimate to be greater than zero, this may justify proceeding to a fully-powered study. We have clarified this in our introduction, aims and discussion:

“No trials have evaluated the impact of supervised, multidisciplinary prehabilitation for people preparing for autologous stem cell transplant… The aim of this pragmatic pilot trial is to explore whether supervised, multidisciplinary prehabilitation for patients preparing for autologous stem cell transplant can positively affect physical capacity after stem cell transplant… This trial will establish estimates of effect to inform a larger definitive trial.”(Page 6, line 125)

“If this trial finds preliminary evidence of efficacy and safety it may inform design of a definitive randomised controlled trial that may transform the stem cell transplant clinical pathway. (Page 16, line 300)

“The sample size for this pilot study is sufficient to estimate whether the assumed clinically important effect of 0.5 is realistic. If we find the estimate to be greater than zero, this may justify proceeding to a fully-powered study [33].” (Page 18, line 329)

Cocks, Kim, and David J. Torgerson. "Sample size calculations for pilot randomized trials: a confidence interval approach." Journal of clinical epidemiology 66.2 (2013): 197-201.

2) Comment: It is not clear to me why the authors decided to measure self-efficacy. I think they need to explain why it is important for this population. The short definition presented in table 2 is not enough to convince the reader of its relevance in this study. They would need to discuss self-efficacy and other outcomes somewhere in the methods section or maybe in the introduction.

Response: Physical activity is recommended as part of standard cancer care in best practice guidelines. A goal of prehabilitation is to establish positive physical activity behaviours to build and maintain physical capacity. It represents a ‘teachable moment’ for behaviour change. Both physical capability and self-efficacy are critical determinants of physical activity behaviour and participation, therefore should be included as an outcome to explore any mediating effect on our outcomes. 

We have added to the introduction to highlight this relationship as follows:

“Prior to stem cell transplant, patients already experience impaired physical function and quality of life [19]. Consequently, patients with haematological malignancies face any barriers to participating in physical activity and want to increase their physical activity behaviors [20]”… It also represents a ‘teachable moment’ whereby patients can gain skills and increase their self-efficacy to be physically active in the long-term [22,23].” (Page 5 line 104)

We have also included a rationale for including self-efficacy in additional file 2

“Self-efficacy will be assessed as it is an important determinant of physical activity behaviour change [25].”

3) Comment: The authors will do a great job recording reasons for non-participation in exercise (or non-adherence to exercise program) which is very important. However, they are planning to report only medical-related reasons, and my question is, why did they ignore psychosocial reasons for non-adherence? I think self-efficacy is also can be a reason for non-adherence as well. It is unnecessary (and not feasible) to report all reasons for non-adherence, but they need to discuss these issues since this is a study protocol paper. They need to be clear about this, and if they are interested only in medical reasons for non-completion or non-adherence, they should mention this explicitly in the methods section.

Response: We agree psychosocial reasons for non-adherence are also important to report and we will be recording these (e.g. reasons such as forgot, work-related). We have now clarified this in the methods section as follows,

“Reasons for non-participation in an exercise session or non-completion of the program, including both medical (e.g. pain, fatigue, unwell) and psychosocial reasons (e.g. work-related, forgot, other appointments), will be recorded.” (Page 13, line 252)

4) Comment: Finally, the validity of the outcome measures should also be discussed in the methods.

Response: We agree the validity of outcome measures should be discussed in the paper. In the interest of word count, we have included a supplementary file (Additional File 2) with this information. 

Reviewer #2: 

1. Comment: The overall goal of this pilot study should be the hypothesis generating instead of hypothesis testing.

Response: We agree. As per our response to reviewer one, this pilot trial is required to explore whether it is possible to implement prehabilitation after autologous stem cell transpant and generate estimates of effect to inform a larger definitive trial. 

We have clarified this in the introduction, aims and discussion as follows:

“No trials have evaluated the impact of supervised, multidisciplinary prehabilitation for people preparing for autologous stem cell transplant… The aim of this pragmatic pilot trial is to explore whether supervised, multidisciplinary prehabilitation for patients preparing for autologous stem cell transplant can positively affect physical capacity after stem cell transplant… This trial will establish estimates of effect to inform a larger definitive trial.”(Page 6, line 125)

“The sample size for this pilot study is sufficient to estimate whether the assumed clinically important effect of 0.5 is realistic. If we find the estimate to be greater than zero, this may justify proceeding to a fully-powered study [33].” (Page 18, line 329)

Cocks, Kim, and David J. Torgerson. "Sample size calculations for pilot randomized trials: a confidence interval approach." Journal of clinical epidemiology 66.2 (2013): 197-201.

2. Comment: The permuted block randomization method was proposed; however, it is unclear if the randomization process includes any stratification factors.

Response: The randomisation process did not include stratification factors due to the small sample size and exploratory nature of the trial. We now include this detail as follows

“…using permuted blocks without stratification.” (Page 7, line 146)

3. Comment: The power analysis is based on the effect size method, which is not very informative. The authors should perform the precision analysis, i.e., the estimated width of the 95% CI, to reassure the proposed study sample size is sufficient. 

Response: Thank you. We have referred to Cocks and Torgerson (2013) to provide a more meaningful justification of the sample size for our pilot study.

“No minimal clinically significant difference has been calculated in patients receiving autologous stem cell transplant therefore it was estimated to be 41 m based on half a standard deviation [31] of scores of a mixed cohort of cancer survivors [32]. For this pilot study, to produce a one-sided 80% confidence limit that would exclude an effect of 0.5 would require a sample size of n=12; and a one-sided 90% confidence interval would require a sample size of n=28. Therefore, we aimed to sample n=22 for this pilot study (Cocks and Torgerson 2013). Approximately 30 people are treated with autologous stem cell transplant at the health service annually. Our sample size represents a recruitment rate of 75% which is similar to a recently completed cancer rehabilitation trial at the health service [33]. (Page 15, line 266)

4. Comment: In addition, please specify the statistical test that was used for the power analysis, and clearly specify if the power analysis was based on the one- or two-sided type I error.

Response: The original sample size estimation was based on a two-sided test with a type 1 error of 0.05. However, on reflecting on your previous comment we have rewritten the sample size estimation according to the recommendations of Cocks and Torgerson (2013) based on the width of the confidence interval, assuming an effect size of 0.5 (see response to comment 3).

5. Comment: The trial has one primary endpoint, but there are almost 10 secondary outcomes in this very small trial. The authors may need to apply the multiple comparison method for the secondary outcomes when reporting the results.

Response: We understand there are divergent views on this issue. However, as this trial is a pilot and exploratory we are not applying a universal hypothesis. That is, we are not testing that the two groups will are the same for all outcomes measured. Therefore consistent with recommendations (Perneger 1998) we do not propose to correct for multiplicity. We have added the following to statistical analysis:

“As this trial is exploratory and there is no universal hypothesis our analysis will not correct for multiplicity [35].” (Page 15, line 282)

Perneger, Thomas V. "What's wrong with Bonferroni adjustments." BMJ 316.7139 (1998): 1236-1238.

6. Comment: The discussion of the patients’ compliance measurement is superficial.

Response: As per reviewer 1, we have now included detail about the inclusion of psychosocial reasons for non-participation. 

“Reasons for non-participation in an exercise session or non-completion of the program, including both medical (e.g. pain, fatigue, unwell) and psychosocial reasons (e.g. work-related, forgot, other appointments), will be recorded.” (Page 13, line 251)

We have also added more detail about what will be included in the nutrition and exercise logs beyond the number and duration of completed sessions as follows:

“The exercise log will record exercise type, intensity, duration, frequency and modifications. The nutrition log will record weight, general health changes, 24-hour food recall and session goals.” (Page 12, line 221)

7. Comment: The authors should clearly discuss the statistical test that will be used for each secondary outcome.

Response: Linear mixed models will be applied for all continuous outcomes including the primary and secondary outcomes. Secondary outcomes related to health service utilisation including number of emergency department, SURC presentations an hospital admissions will be analysed using a negative binomial regression model and reported as an incidence rate ratio. This detail is outlined as follows:

“The primary outcome (physical capacity at 4-weeks post-transplant) will be analysed using linear mixed effects models… A similar approach (linear mixed effects model) will be used for analysis of continuous secondary outcomes collected longitudinally… The number of emergency department, SURC presentations and hospital admissions will be reported as an incidence rate ratio using a negative binomial regression model.” (Page 15, line 281)

8. Comment: The authors should clearly specify the methods that will be used to estimate the 95% Confidence Intervals (CIs).

Response: 95% confidence intervals will be generated as an output from our statistics software (IBM SPSS version 28) when the linear mixed models are applied and will be reported in line with best practice in the main publication when results are available. 

9. Comment: The authors should discuss the statistical software that will be used to analyze the data.

Response: Thank you for identifying this oversight. Data will be analysed using IBM SPSS statistics version 28.

“Data will be analysed using IBM SPSS version 28.” (Page 16, line 290)

---

## [Decision Letter · Decision Letter 1]

3 Nov 2022

Prehabilitation to Improve outcomes afteR Autologous sTem cEll transplantation (PIRATE): A pilot randomised controlled trial protocol

PONE-D-21-32897R1

Dear Dr. Dennett,

We’re pleased to inform you that your manuscript has been judged scientifically suitable for publication and will be formally accepted for publication once it meets all outstanding technical requirements.

Within one week, you’ll receive an e-mail detailing the required amendments. When these have been addressed, you’ll receive a formal acceptance letter and your manuscript will be scheduled for publication. Please note that I've asked my colleagues in the editorial office to request the original trial protocol submitted to your IRB, as this is required for all clinical trials and clinical trial study protocols. 

Kind regards,

Dario

Dario Ummarino, PhD

Senior Editor

PLOS ONE

Reviewers' comments:

Reviewer's Responses to Questions

**Comments to the Author**

1. Does the manuscript provide a valid rationale for the proposed study, with clearly identified and justified research questions?

Reviewer #1: Yes

Reviewer #2: Yes

2. Is the protocol technically sound and planned in a manner that will lead to a meaningful outcome and allow testing the stated hypotheses?

Reviewer #1: Yes

Reviewer #2: Yes

3. Is the methodology feasible and described in sufficient detail to allow the work to be replicable?

Reviewer #1: Yes

Reviewer #2: Yes

4. Have the authors described where all data underlying the findings will be made available when the study is complete?

Reviewer #1: No

Reviewer #2: Yes

5. Is the manuscript presented in an intelligible fashion and written in standard English?

Reviewer #1: Yes

Reviewer #2: Yes

6. Review Comments to the Author

You may also provide optional suggestions and comments to authors that they might find helpful in planning their study.

Reviewer #1: Dear authors,

Thank you for carefully addressing my comments. I am happy with the revisions and I do not have anything else to add.

I wish you all the best,

Reviewer #2: The authors have responded well to the statistical issues raised in the previous review. There is no further statistical concern about this revised manuscript.

7. PLOS authors have the option to publish the peer review history of their article (what does this mean?). If published, this will include your full peer review and any attached files.

Reviewer #1: No

Reviewer #2: No

---

## [Editor Report · Acceptance letter]

19 Apr 2023

PONE-D-21-32897R1 

Prehabilitation to Improve outcomes afteR Autologous sTem cEll transplantation (PIRATE): A pilot randomised controlled trial protocol 

Dear Dr. Dennett:

I'm pleased to inform you that your manuscript has been deemed suitable for publication in PLOS ONE. Congratulations! Your manuscript is now with our production department. 

Kind regards, 

on behalf of

Dr Dario Ummarino, PhD 

Staff Editor

PLOS ONE